# The Ketogenic Diet in the Prevention of Migraines in the Elderly

**DOI:** 10.3390/nu15234998

**Published:** 2023-12-02

**Authors:** Michal Fila, Jan Chojnacki, Elzbieta Pawlowska, Piotr Sobczuk, Cezary Chojnacki, Janusz Blasiak

**Affiliations:** 1Department of Developmental Neurology and Epileptology, Polish Mother’s Memorial Hospital—Research Institute, 93-338 Lodz, Poland; michal.fila@iczmp.edu.pl; 2Department of Clinical Nutrition and Gastroenterological Diagnostics, Medical University of Lodz, 90-647 Lodz, Poland; jan.chojnacki@umed.lodz.pl (J.C.), cezary.chojnacki@umed.lodz.pl (C.C.); 3Department of Pediatric Dentistry, Medical University of Lodz, 92-217 Lodz, Poland; elzbieta.pawlowska@umed.lodz.pl; 4Emergency Medicine and Disaster Medicine Department, Medical University of Lodz, 92-209 Lodz, Poland; piotr.sobczuk@umed.lodz.pl; 5Department of Orthopaedics and Traumatology, Polish Mother’s Memorial Hospital—Research Institute, Rzgowska 281, 93-338 Lodz, Poland; 6Faculty of Medicine, Collegium Medicum, The Mazovian Academy in Plock, 09-402 Plock, Poland

**Keywords:** migraine, headache, ketogenic diet, elderly, ketone bodies, β-hydroxybutyrate

## Abstract

Migraines display atypical age dependence, as the peak of their prevalence occurs between the ages of 20–40 years. With age, headache attacks occur less frequently and are characterized by a lower amplitude. However, both diagnosis and therapy of migraines in the elderly are challenging due to multiple comorbidities and polypharmacy. Dietary components and eating habits are migraine triggers; therefore, nutrition is a main target in migraine prevention. Several kinds of diets were proposed to prevent migraines, but none are commonly accepted due to inconsistent results obtained in different studies. The ketogenic diet is featured by very low-carbohydrate and high-fat contents. It may replace glucose with ketone bodies as the primary source of energy production. The ketogenic diet and the actions of ketone bodies are considered beneficial in several aspects of health, including migraine prevention, but studies on the ketogenic diet in migraines are not standardized and poorly evidenced. Apart from papers claiming beneficial effects of the ketogenic diet in migraines, several studies have reported that increased levels of ketone bodies may be associated with all-cause and incident heart failure mortality in older adults and are supported by research on mice showing that the ketogenic diets and diet supplementation with a human ketone body precursor may cause life span shortening. Therefore, despite reports showing a beneficial effect of the ketogenic diet in migraines, such a diet requires further studies, including clinical trials, to verify whether it should be recommended in older adults with migraines.

## 1. Introduction

Nutrition in the elderly is an important aspect of their lives, as during the life course, some products that are profitable or neutral become either non-tolerable or harmful. It is especially important for many age-related diseases when some dietary components are risk factors for a particular disease.

Migraines belong to the most disabling disorders worldwide. It has a specific age-related characteristic, as its prevalence decreases with age, and migraine headaches are less frequent and of lower intensity in older adults [1]. However, the prevalence of migraines in the elderly is still high, and headaches are a common problem in aged individuals seeking medical care due to cognitive decline [2].

Certain dietary components are among the most consistently reported triggers for migraine attacks [3]. Several studies have claimed that food and drinks are the most often indicated migraine headache triggers [4]. Alcoholic beverages, caffeine, milk, chocolate, and cheese have been reliably reported as migraine triggers. Moreover, some eating habits and dietary patterns are associated with migraines [5]. In particular, eating too much or too less may be positively correlated with migraine attacks [6]. Nighttime snacking and having late dinners were considered as factors that may reduce migraine attacks [7]. However, it should be considered that withdrawal of some components of the everyday diet that may be migraine triggers may provoke migraine attacks, and alcohol and caffeine are examples of such substances [8]. 

Although it has been accepted for years that saturated fats elevate “bad” cholesterol, and as a consequence, many anti-cholesterol preventive strategies, including “healthy” diets, have been developed, we observe an increase in coronary heart disease incidence and prevalence [9]. However, reduction in the consumption of saturated fats is often associated with increased consumption of carbohydrates that are broken down into sugars and contribute to obesity, diabetes, and heart diseases [10]. The ketogenic diet reduces the total carbohydrate and protein intake and increases fats. Although the beneficial effects of the ketogenic diet have been reported in many studies, there is no scientific evidence of the prominence, safety, and sustainability of the ketogenic diet [11].

Although several kinds of diets, including the ketogenic diet, have been proposed to exhibit a beneficial potential for migraines, there is no consensus on any specific diet as an “anti-migraine” diet. This is likely due to the differences in the results obtained in different studies. Therefore, it is not clear whether the ketogenic diet is effective in migraine prevention. The other question is whether it can be safely applied in elderly individuals suffering from migraines.

In this narrative/perspective review, we present information on migraines and their relation to nutrition, the ketogenic diet, and its application in migraines and elderly individuals. Finally, we present arguments against the application of the ketogenic diet in the elderly suffering from migraines. 

Migraines are one of the most serious reasons for disability and affect a significant part of the older adult population, adding another health concern in this group of individuals. The ketogenic diet has been recommended for migraine prevention, but there is not a clinical trial evaluating its efficacy and safety in elderly migraine sufferers. This review has been written to provide information and arguments to verify whether the ketogenic diet should be applied in the prevention of migraines in the elderly. It is obvious that the final conclusion must be verified empirically through pre-clinical and clinical studies.

## 2. Migraines and Their Relation to Age and Nutrition

Migraines, constituting a neurological disease, are ranked as the second (first in young women) cause of disability worldwide [12]. It is not easy to estimate the global prevalence of migraines, but as suggested lastly by Steiner and Stovner, its best estimate is 14–15%, and migraines are responsible for 4.9% of worldwide ill health quantified in years lived with disability [13].

The exact mechanisms underlying migraine pathogenesis are poorly known, and the main reason for that is restricted access to the human target material and limited suitability of cellular and animal models of human migraines. Despite these difficulties, many migraine cases can be successfully treated using antibodies against calcitonin gene-related peptide (CGRP) and its receptor and agonists of the CGRP receptor [14]. In fact, introducing these compounds in the market revolutionized migraine treatment [15]. 

Most diseases increase their prevalence with age, but migraines do not follow that rule. For a great majority of patients, the most intense migraine-related headaches occur between 20 and 40 years old [16] (Figure 1). Therefore, migraines constitute a disease that mostly affects people at their most productive age. We recently hypothesized that this may be associated with the stress linked with final exams, job hunting, maternity/paternity, intense work, and other aspects of social stress [17].

The migraine prevalence in women is reported up to three times higher than in men. Although the exact reason for this diversity is not known, several possible mechanisms are considered to underline this difference, including sex hormones, pregnancy, the menstrual cycle, X-linked form of migraines, and mitochondrial transmission [18,19].

The International Classification of Headache Disorders (ICHD-3) distinguishes many categories of migraines, including migraines without and with aura [20]. Aura occurs in about 25% of migraine cases, and usually takes place before headaches [21]. Typical aura symptoms are visual and sensory, but several other aspects of aura have been reported. Migraines with aura are further divided into several categories, including abdominal migraines, hemiplegic migraines, retinal migraines, migraines with brainstem aura, and others [20]. Headaches in symptomatic migraines are, per definition, linked with their causes, but in others, headaches are induced by migraine triggers. Several dietary components, as well as dietary patterns and eating habits, are among the most consistently reported migraine triggers [22]. Moreover, it has been hypothesized that this may be a mutual relationship, i.e., some foods may induce migraine headaches, but migraines may also alter some eating habits and diet composition, and this is not very surprising, as such a situation occurs in many other diseases [23].

In contrast to nutritional migraine triggers, some dietary components may exert a beneficial effect in migraines, as they may prevent headache attacks and lower their intensity [24]. These components may meet an increased energy demand by the migraine-affected brain [25]. Several nutrients were suggested to improve brain energy balance in migraine attacks [24]. Brain energy is produced from ATP synthesized in brain mitochondria, and we have suggested several nutrients that may improve brain energy in migraines [24], namely thiamine, lipoic acid, riboflavin, pyridoxine, magnesium ions, cobalamin, folate, niacin, coenzyme Q10, carnitine, and melatonin, but the context in which they are administrated is also important.

Furthermore, CGRP, which is at the center of migraine pathogenesis, may induce anorexigenic neuropeptides and inhibit orexigenic neuropeptides, and in this way, regulate appetite and satiety in rats [26,27]. Therefore, nutrition and migraines have a strong connection, and it is not surprising that certain kinds of dietary interventions have been proposed to prevent migraines, and the ketogenic diet is among them.

## 3. Migraines in the Elderly

In the elderly, both the prevalence and incidence of migraines decrease with age, and migraine attacks in aged individuals are milder than in their younger counterparts [28]. The clinical presentation of migraines in the elderly is different than in younger persons, which complicates its diagnosis. Migraine therapy in the elderly is complex, not only due to the different clinical picture but also to the significantly high number of comorbidities and polypharmacy in this age group. The number of clinical trials and experimental studies in aged migraine patients is relatively low, which impedes progress in studies aimed at exploring mechanisms of migraine pathogenesis. These elements have decided that migraines in the elderly are more undiagnosed and untreated than in younger individuals. 

Elderly patients usually present bilateral headaches, while younger patients present hemispheric ones. They may also present aura only without headaches. Migraine attacks in this group of patients may be shorter and less intense and largely free of migraine-associated symptoms, such as nausea, vomiting, and photo/phonophobia [2,29,30]. Cognitive impairments and reversible loss of memory occur more frequently in elderly migraine patients than in their younger counterparts [2,31]. Like in the general population, headaches in the elderly occur more frequently in women than in men. Due to their general physical and psychic conditions, migraines may present a higher burden for the elderly than for younger individuals. 

Late-life migraine accompaniments (LLMAs) is a term describing “transient neurological episodes in older individuals that mimic transient ischemic attacks” [32,33]. LLMAs are more common in the elderly [34,35]. LLMAs include visual, paresthesia, and speech disturbances and paresis and occur suddenly before, during, or after a migraine attack [34]. Such a high prevalence of LLMAs in the elderly implies that this group of patients may be affected by the most serious syndromes associated with migraines, including ischemic stroke, at a higher rate than younger patients [36].

The elderly are characterized by a high prevalence of serious symptoms associated with age, including hypertension, diabetes, and cerebrovascular diseases, that require medications that may be in conflict with migraine treatments. Moreover, some symptoms typical for the elderly, including reduced liver mass and blood flow, as well as reduced renal mass and globular filtration rate, constitute pharmacokinetic and pharmacodynamic problems for pharmacological migraine treatment. Medication overuse, and consequently, medication overuse headaches (MOHs), are common in this group of patients and may be as high as 30% [37].

As mentioned, migraine treatment in the elderly is more challenging than in younger patients due to many comorbidities typical for that age and polypharmacy [38]. However, several studies have suggested that polypharmacy in young adults suffering from headaches is comparable with that in the elderly, but the drugs used by younger individuals mainly target headaches and rarely comorbidities [39]. 

Non-pharmacological preventive treatments of migraines, such as sleep hygiene, outdoor walking, following a meal schedule, and proper hydration, may be easier to follow for the elderly, especially those retired, than younger individuals. Acute pharmacological treatment in the elderly migraine sufferers with simple analgesic, NSAIDs, triptans, and others should be recommended with care due to their possible interactions with other medications used in comorbidities. Long-term preventive treatments, including anti-CGRP therapy, for the elderly still wait for a definite conclusion from clinical and preclinical studies.

In summary, migraines in the elderly occur at a lower prevalence than in younger individuals, and headache attacks occur at a lower intensity (Figure 2). It is mostly specified by bilateral headaches, and is more frequently associated with transient cognitive and memory impairments. It is also typified by serious syndromes related to ischemic stroke. However, it constitutes a complex diagnostic and therapeutic problem, as its phenotype is affected by comorbidities, as well as those with secondary headaches. Migraine treatment in this group of patients is additionally complicated by polypharmacy, often leading to MOHs. 

## 4. The Ketogenic Diet

Although historically, the ketogenic diet (keto diet) was aimed at treating epilepsy, nowadays it is thought to help weight loss and consequently fight obesity-related diseases, including type 2 diabetes mellitus, hyperlipidemia, heart disease, and cancer [11,40,41].

A ketogenic diet is basically characterized by a high intake of fat, moderate protein intake, and low intake of carbohydrates. Typically, it consists of 70–80% of fats, 15–20% of proteins, and 5–10% of carbohydrates [41].

The limitation in carbohydrate intake may result in decreased insulin secretion, a transition into a catabolic state, and the induction of gluconeogenesis and ketogenesis [42,43]. Gluconeogenesis leads to the internal production of glucose, mainly in the liver, from pyruvate, lactic acid, glycerol, and glucogenic amino acids, such as alanine, arginine, asparagine, cysteine, glutamine, aspartic acid, glutamic acid, and glycine [44]. However, if the internal production of glucose is not sufficient to provide the necessary amount of ATP, the metabolic pathway switches to ketogenesis, which produces ketone bodies to replace glucose as the main supply of energy. Ketone bodies, including β-hydroxybutyrate, acetoacetate, and acetone, are synthesized in the liver and then oxidized outside the liver (Figure 3). They are transformed into acetyl coenzyme A in mitochondria and enter the tricarboxylic acid cycle and oxidative phosphorylation to produce ATP [45]. 

Low insulin secretion during ketogenesis results in decreased fat and glucose accumulation, which, along with other changes, leads to increased breakdown of fat, resulting in fatty acid production [45]. When the ketogenic diet is applied, the accumulation of ketone bodies is sustained, and the organism reaches the state of nutritional ketosis, which is generally considered as safe and may be beneficial for the organism [41]. However, the overproduction of ketone bodies may lead to ketoacidosis, which is a life-threatening condition leading to blood acidosis [46]. Ketone bodies produce more ATP than equivalent amounts of glucose, e.g., 100 g of glucose yields 8.7 kg of ATP, while the same amount of acetoacetate, a primary product of fatty acid metabolites, yields 9.4 kg of ATP [47].

In the context of migraines, it is important that ketone bodies can penetrate the blood–brain barrier to provide energy to the brain, whose deficit is typical for migraines [48]. Such an energy deficit may be a source of reactive oxygen and nitrogen species (RONS) that may damage macromolecules and brain structures important in migraine pathogenesis [25]. In turn, ketone bodies are reported to have antioxidant properties [49].

Ketone bodies are transported into the brain through monocarboxylate transporters (MCTs) that are independent of insulin and glucose transporters (GLUTs), including GLUT3, which is a main glucose transporter in neuronal and glial cells [50]. On the other hand, ketone bodies may upregulate MCTs [51]. However, ketone bodies can be also endogenously produced in the brain via astrocytes that are able to metabolize free fatty acids and the ketogenic amino acids lysine and leucine [52].

Ketone bodies are not the only “superfuel” as virtually all brain cells can use ketone bodies as substrates for energy production, but they may also play a signaling role [53,54]. Both of these roles are important in migraine pathogenesis [55]. In general, many mechanistic pathways may underline the migraine-related effects induced by the ketogenic diet, including metabolism, neurotransmission, synaptic recycling, ion channels, autophagy, mitochondrial quality control, the gut microbiome, the endocrine system, epigenetics, and the immune system (reviewed by the authors of [56]).

The ketogenic diet stimulates mitochondrial metabolism, which is mostly evidenced in the muscles and is enhanced by physical exercise [57]. However, it was shown that the ketogenic diet in rats decreased mitochondrial biogenesis and reduced mitochondrial respiration [58]. Mechanistically, increased levels of β-hydroxybutyrate, a histone deacetylase 2 (HDAC2) inhibitor, promoted the histone acetylation of the promoter of the *SIRT7* promoter, activating its transcription, which inhibited the transcription of mitochondrial ribosome-encoding genes and mitochondrial biogenesis. Therefore, mitochondria are an important element in the action of ketone bodies.

In general, the ketogenic diet has been often considered as beneficial or at least safe for health. However, a number of recent studies have suggested a change in this paradigm. These studies will be presented in subsequent sections.

## 5. The Ketogenic Diet and Ketone Bodies in Migraines

The International Ketogenic Diet Study Group lists migraines as a neurological disease that can be potentially beneficially targeted via the ketogenic diet [59]. 

Migraines are associated with increased energy demands and decreased energy production by the migraine-affected brain [25]. Therefore, if the source of energy is limited, e.g., by low levels of carbohydrates typical for the ketogenic diet, this additionally deepens the energetic crisis typical for the brain in migraines. Therefore, how may the ketogenic diet exert a beneficial effect in migraines? The answer may not only lie in a switch between the utilization of glucose and ketone bodies as sources of energy but also signaling functions of ketone bodies. 

However, independently of a source, energy production in the brain occurs in mitochondria, so they stand in a central position in the energetic aspect of migraine pathogenesis. The role of mitochondrial biogenesis and functioning in migraine pathogenesis has been debated for years, but so far, a definitive conclusion has not been drawn. Mitochondria, due to their role in generating energy and in the production of RONS, are involved in many physiological and pathological conditions; thus, attributing a significant role of mitochondria to migraine pathogenesis exclusively due to energy crises and oxidative stress would be a truism, important in many other pathologies. As mentioned, mitochondrial transmission may contribute to up to a three times higher prevalence of migraines in women than in men, although studies on migraine-specific loci in mtDNA are inconclusive [60].

The strongest support for the role of mitochondria in migraine pathogenesis likely comes from the association between mitochondrial diseases and migraines [61]. Stratification of migraine cases into specific types of mitochondrial diseases implied that migraines may not be just an association syndrome in a specific mitochondrial disease but, instead, might express a universal vulnerability of the CNS to some factors associated with mitochondrial dysfunction accompanying that disease. Riboflavin (vitamin B2) is a central component of flavin mononucleotide and flavin adenine dinucleotide, which are important in critical mitochondrial processes, including the metabolism of amino acids, purines, and fatty acids [62]. On the other hand, riboflavin was reported to show a preventive action against migraines in 8 randomized controlled clinical trials and 1 clinical trial with 673 subjects [63]. Riboflavin, similar to other B vitamins, is needed for the breaking down and processing of ketone bodies [64]. In turn, ketone bodies may improve mitochondrial functions through different mechanisms, including the attenuation of an increased level of H3_K27me2K36me1 and mitochondrial dysfunction followed by repressed peroxisome proliferator activator gamma coactivator 1 alpha (PGC-1α) downregulation [65]. In another study, a decreased level of mRNA expression of the *PPARGCA1* gene encoding the PGC-1α protein along with fewer mtDNA copies were observed in a rat model of migraines [66]. Therefore, mitochondria can present an important link between the ketogenic diet and migraines, and PGC-1α may play a mechanistic role in this link.

Oxidative stress is involved in the pathogenesis of many disorders, but for a definite majority of them, the precise mechanism of that involvement is not known. The role of oxidative stress in migraine pathogenesis may be underlined by several mechanisms, including energy generation in the brain to amend energy deficits typical for migraine-affected brains, neuroinflammation, impaired DNA damage response, and others [67]. The ketogenic diet was reported to ameliorate oxidative stress and improve mitochondrial functions in rats with traumatic brain injury [68]. The former effect was mainly underlined by the overexpression of the antioxidant enzymes nicotinamide adenine dinucleotide (NADH), dehydrogenase quinone 1 (NQO1) and superoxide dismutase 1/2 (SOD1/2), whereas the latter effect was indicated by an increase in the activity of mitochondrial respiratory complexes II and III. In a later work, it was shown that the ketones β-hydroxybutyrate and acetoacetate exerted a beneficial effect on rat mitochondria isolated from neocortical neurons and exposed to a high concentration of calcium [69]. A combination of both ketone bodies decreased the death of the neurons and prevented alterations in the neur onal membrane induced by glutamate. Both ketones decreased the production of mitochondrial RONS and increased NADH oxidation in the mitochondrial respiratory chain. Therefore, ketone bodies may decrease RONS formation in neurons subjected to glutamate excitotoxicity by raising the NAD+/NADH ratio and increasing mitochondrial respiration in neocortical neurons. This mechanism may be important in migraine protection by ketones through the restoration of bioenergetic functions challenged by oxidative stress [25,70,71].

Cortical spreading depolarization (CSD) is a wave of neurons and neuroglia depolarization within the cerebral cortex and is attributed to migraine aura [72]. A protective effect of a ketogenic diet enriched with middle- and long-chain triglycerides on the CSD propagation was observed in rats, reflecting a decrease in brain cerebral excitability, typical for migraines [72]. 

There are several papers that have dealt with ketogenic therapies to prevent migraines or ameliorate headache attacks (reviewed in [59]). However, these ketogenic therapies are not standardized, and it is difficult to compare their results. The ketogenic diet we described in the previous sections can be considered as a “classical ketogenic diet”, typically with the 3–4:1 ketogenic ratio (3–4 mass units of fat per 1 mass unit of protein and carbohydrate). In addition to this, the low-calorie ketogenic diet, the modified Atkins diet, and a diet enriched with exogenous source(s) of β-hydroxybutyrate are considered dietary interventions that induce ketosis. Several randomized clinical trials evaluating the role of ketogenic interventions in migraines were performed. The first such trial by Di Lorenzo et al. included overweight migraine patients during a weight loss intervention [73]. Therefore, these results cannot be directly related to this review, as (1) migraines and overweight/obesity are important and autonomous issues, and (2) obesity in the elderly is a problem, but it is somehow overwhelmed by unexplained weight loss, which is typical for this group of patients. A recent review by Neri et al. presented clinical trials and other studies of migraine patients with ketogenic interventions resulting in the production of endogenous ketone bodies or the administration of exogenous ketone bodies [74]. That review also pointed at the reduction in the number of attacks and their severity in most clinical trials on the ketogenic diet in migraines [74]. Moreover, only a few studies reported mild side effects, but not all of them addressed that issue. However, the authors pointed that many studies were of moderate-to-low quality and reported inconsistent results, resulting in a poor recommendation strength. A low-to-moderate risk of bias was observed in most studies. So far, no clinical trials with a ketogenic intervention in elderly migraine patients have been performed.

## 6. The Ketogenic Diet and Ketone Bodies in Aging 

There are conflicting opinions on the ability of older people to change their life-long habits to implement health recommendations. One says that they may have difficulties adapting to a new lifestyle; the other claims that they are more open to lifestyle modifications than their younger counterparts [75,76]. A study aimed to assess whether older patients, defined as 65 years of age or older, were able to adopt and maintain a ketogenic diet in a cohort of US citizens [77]. In 67% of patients, the ketogenic diet resulted in accomplishing its goals, concerning weight loss, glucose control, and anticancer effects. Adverse effects, mainly dyslipidemia and constipation, were reported by 15% of patients. These studies showed that older patients, who suffer from various health problems, may adopt to the ketogenic diet, and such a diet may effectively and positively change their health status. It should be underlined that most patients in that study had notable baseline morbidity.

Caloric restriction and resulting ketogenesis are the most consistently reported factors that may prolong life span [78]. It is important that these factors are reported to not only extend longevity but also health span in adult humans and mice [79,80]. However, both caloric restriction and the ketogenic diet are dietary interventions that may cause multiple metabolic changes, and it is difficult to judge about their mechanisms beyond their beneficial action [81]. 

Two human studies reported a harmful effect of ketone bodies. A positive correlation between circulating ketone bodies and all-cause mortality was observed in the general population-based cohort study Prevention of Renal and Vascular Endstage Disease Intervention (PREVEND) in individuals aged 54 ± 12 years [82]. This study also revealed that an increased all-cause mortality was associated with an increased fatty liver index, a proxy for non-alcoholic fatty liver disease, and this increase was partly mediated by circulating ketone bodies. A recent study reported a dose–response relationship of a 50% increase in all-cause mortality between the lowest and highest quintiles of ketone body concentrations in White and Black Americans [83]. Moreover, the levels of ketone bodies were associated with incident heart failure and higher all-cause mortality. These associations suggest a potential detrimental effect of ketone body metabolism in aging.

The mechanism behind these observations in humans has been partly explained in an experiment with homozygotic mice with a mutation in the 3-hydroxy-3-methylglutaryl-CoA synthase 2 (HMGCS2) gene encoding for the protein critical for the second stage of ketogenesis [84,85]. These animals are incapable of endogenous ketogenesis. The HMGCS2^−/−^ mice were given a normal diet, a ketogenic diet (comprising 4.5% carbohydrates, 80.8% fats, and 14.7% proteins), and a normal diet supplemented with 1,3-butanediol, a precursor of β-hydroxybutyrate, which showed that ketone bodies may exert a significant effect on the life span of animals in two periods of life: just after the birth and older age. Specifically, mice on a ketogenic diet displayed a higher level of mortality in old age, and mice on a diet supplemented with 1,3-butanediol exhibited higher mid-life mortality, but their mortality in old age was like that of control animals. An ad libitum ketogenic diet increased mortality, suggesting that not only the proportion of the main components of the ketogenic diet but also its rate of administration may determine its outcome. 

Although traumatic headaches represent a different class of headaches than those typical for migraines, ketone bodies have therapeutic potential in traumatic brain injury, as shown in experimental animals [86]. However, the question of the age dependence of such a therapy belongs to the most important concerns associated with such a kind of therapy in humans.

Autophagy, a process of removing damaged and no longer needed cellular components with their possible recycling and reuse, plays a fundamental role in metabolism and is closely related to diet and aging [87]. Such a form of autophagy is associated with lysosomal degradation and is called degradative autophagy. However, several cellular components are not needed in their origin and are transported outside the cell in a process called secretory autophagy [88]. Normal autophagy is needed for homeostasis, and many disorders are associated with impaired autophagy [89]. In general, compromised autophagy is considered as a hallmark of aging [90]. On the other hand, several studies with long-lived organisms showed that delayed aging was associated with increased autophagy [91,92,93,94]. Recently, we suggested that an interplay between secretory autophagy in microglia and degradative autophagy in neurons may play a role in migraine pathogenesis with the involvement of brain-derived neurotrophic factor (BDNF) and the interaction of ATP with the purinergic receptor P2X7 (P2X7R) [95]. It was observed that a ketogenic diet upregulated hepatic autophagy in mice [96]. The same authors showed that the ketogenic diet upregulated the autophagosome-associated protein LC3-II in mouse hippocampal and cerebrocortical samples [97]. In general, these studies confirmed further research showing that ketosis may promote brain autophagy by activating Sirtuin 1 and hypoxia-inducible factor-1 [98]. Although vasodilation is no longer considered the sole cause of migraines, the ability of vasoactive substances to induce migraines, the efficacy of drugs targeting vasoactive sites in the brain, and the positive correlation of migraines with cardiovascular diseases make it a causative component in the pathogenesis of migraines [99]. It was shown that mice fed with a high-fat diet exhibited defects in metabolism and autophagy, which were ameliorated with a ketogenic amino acid (KAA) replacement diet [100]. That study showed that the biosynthesis of β-hydroxybutyrate could be impaired via the inhibition of autophagy. Consequently, β-hydroxybutyrate induced a potent vasodilator effect via potassium channels. Finally, it was observed that the prolonged consumption of a high-salt diet negatively regulated both β-hydroxybutyrate biosynthesis and hepatic autophagy, and that resuming of β-hydroxybutyrate bioavailability inhibited high-salt diet-induced endothelial dysfunction. Therefore, these studies suggest a direct mechanism by which ketogenic dietary interventions ameliorate vascular health, which may play a role in the beneficial effect of a ketogenic diet on migraines. The relationship between autophagy and aging suggests that ketogenic interventions in migraines in the elderly may not be sufficient. 

In light of recent research, supplementation of the diet with ketone bodies may result in different effects depending on the timing and the method of such supplementation. The next issue that could be addressed is whether the supplementation occurs in normal or damaged organisms, as the ketogenic diet mat play an important role in tissue repair [82,101]. Another factor that should be included in the consideration of the role of ketogenesis in aging is sex, as it was shown that the difference in quintiles of ketone bodies between women and men was at the border of significance [81]. The ketogenic diet as a dietary intervention should be controlled both quantitively and qualitatively, as its ad libitum form may lead to an uncontrolled rise in ketone bodies.

## 7. Conclusions and Perspectives

The ketogenic diet has been commonly considered beneficial in many health issues, including weight and glucose control, with few adverse effects, good adherence, and good tolerance to it in older adults. The main problem with the recommendation of the ketogenic diet is that it is difficult to conclude about its health consequences due to its different versions and different effects of its essential components. Therefore, the consequences of the ketogenic diet may not be limited to the consequences of the action of increased levels of ketogenic bodies.

The assessment of the health consequences of the ketogenic diet assumes that it enforces the use of ketone bodies instead of glucose as the main source of energy production. However, increased production of ketone bodies is not limited to increased energy production, as they play many important functions in organisms, including signaling, epigenetic regulation, autophagy, and others. 

The ketogenic diet is likely the only diet that is recommended for migraines, but studies performed so far are not standardized to allow a comparison between different outcomes. Moreover, many of these studies are criticized due to their weak evidence, problems with the control group, small sample size, and risk bias. Moreover, there are neither clinical trials nor case-control studies presenting research on the ketogenic diet in an older population. Therefore, currently, there is not enough evidence to recommend the ketogenic diet in the elderly suffering from migraines.

There are some outstanding questions to be answered. Does the specific kind of product constituting the ketogenic diet matter, e.g., what should be the right proportion between products of animal and plant origin? What is the right timing of the onset of the ketogenic diet? Should the ketogenic diet be adjusted to sex? Should the specific characteristics of migraines, e.g., aura presence/absence, abdominal migraines, frequency of attacks, nausea/vomiting, etc., be considered in ketogenic diet recommendations? As long as these and other questions are answered, the ketogenic diet should not be recommended in older individuals with migraines.

Recent works showing the detrimental consequences of an increased concentration of ketone bodies in older individuals, supported by studies on mice fed with a ketogenic diet or a diet supplemented with a precursor of β-hydroxybutyrate, strengthen the conclusion of not recommending the ketogenic diet in older individuals with migraines.

## Figures and Tables

**Figure 1 nutrients-15-04998-f001:**
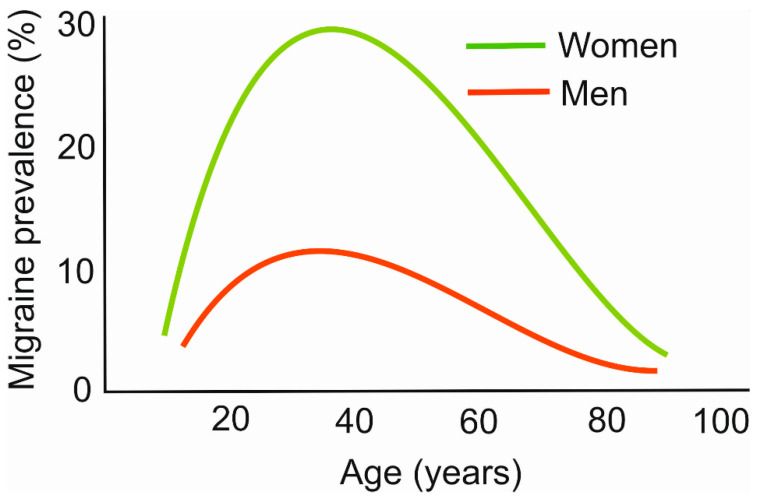
Age dependence of migraine prevalence. The plots are only illustrative and were constructed based on data collected from various sources.

**Figure 2 nutrients-15-04998-f002:**
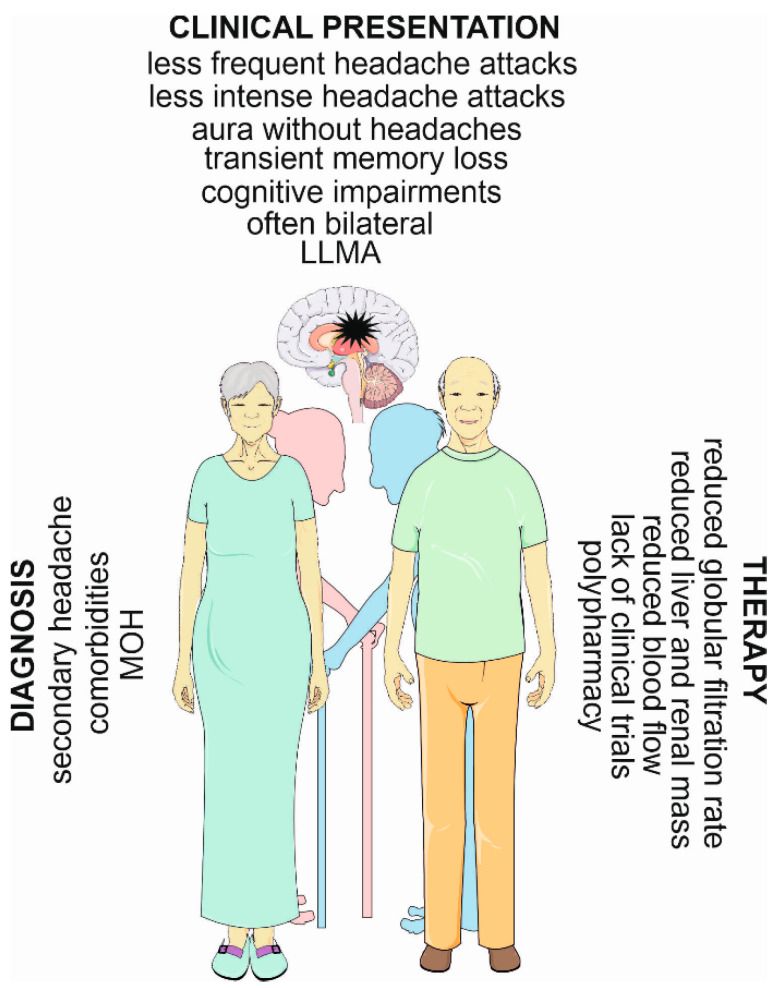
Migraines in the elderly—main diagnostic and therapeutic challenges and distinct symptoms. A migraine has been symbolized here as a black star in the brain. LLMA—late-life migraine accompaniment; MOH—medication overuse headache. Parts of this figure were drawn using pictures from Servier Medical Art. Servier Medical Art by Servier is licensed under a Creative Commons Attribution 3.0 Unported License (https://creativecommons.org/licenses/by/3.0/ (accesesed on 1 November 2023)).

**Figure 3 nutrients-15-04998-f003:**
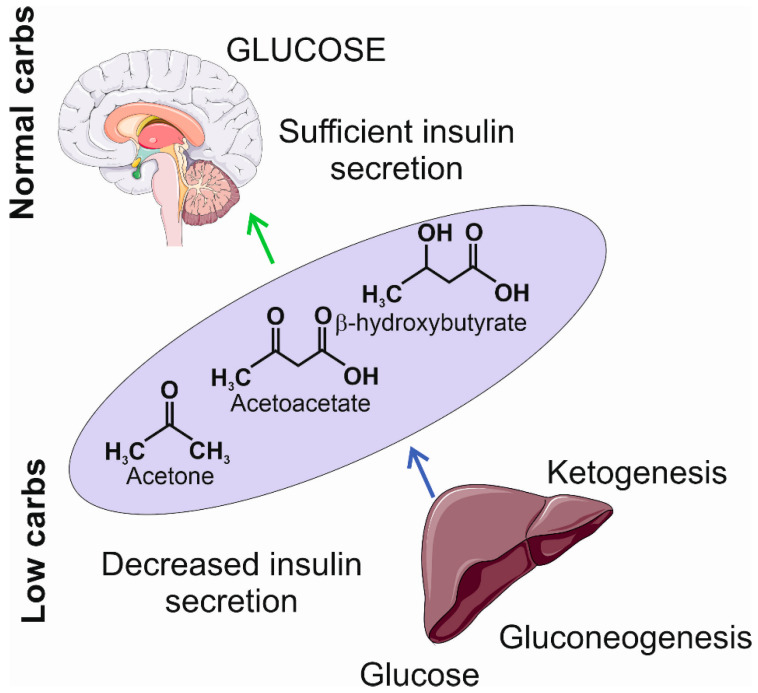
The insufficient levels of carbohydrates induce a decrease in insulin secretion and the activation of gluconeogenesis and ketogenesis. The former produces glucose at a low level; the latter produces ketogenic bodies, including acetone, acetoacetate, and β-hydroxybutyrate, that may become an additional source of energy in the brain. The light violet ellipse symbolizes circulation from which ketone bodies penetrate the brain as they can cross the blood–brain barrier. Parts of this figure were drawn using pictures from Servier Medical Art. Servier Medical Art by Servier is licensed under a Creative Commons Attribution 3.0 Unported License (https://creativecommons.org/licenses/by/3.0/ (accessed on 1 November 2023)).

## Data Availability

Not applicable.

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
