# Peer review of "The Ketogenic Diet in the Prevention of Migraines in the Elderly"

_nutrients, 2023, doi:10.3390/nu15234998_

Round 1

Reviewer 1 Report

Comments and Suggestions for Authors

This is a comprehensive review of the clinical aspects of migraine in the elderly and on the rationale for using ketogenic diet as a treatment in migraine.

I have 2 major comments and suggestions.
1) in § 5 the authors cite only reviews on ketogenic diet in migraine (not mentioning for example the 1st RCT by Di Lorenzo's group) and at the same time underline the heterogeneity of dietary programs. This is obviously important for the effects and side effects of ketogenic diets. I would suggest a table mentioning the major controlled trials underlining their difference in composition.
Moreover in § 5 (but also elsewhere) the reference numbers are not adequate: for instance ref 73 and 74 should be 74 and 75. I recommend to check all references in the manuscript.

2) in § 6 the authors discuss the potential harm ketosis could produce in the elderly. This section is difficult to understand because it is a mixture of experimental data in animals and clinical human data. These 2 data sets should be clearly separated and again it would be instructive and reader-friendly to have a table showing the most prominent animal data on the one hand, and clinical studies on the other. At least the latter should be summarized in a table as there seem to be many contradictory results in the literature (as also mentioned by the authors)

The abstract contrast with §6, because it only mentions the harm ketogenic diet may cause, and not the fact that results are contradictory and that the conclusion that "ketogenic diet should not be recommended in older adults with migraine" could be somewhat nuanced. 

Comments on the Quality of English Language

some typos

Author Response

Reviewer #1

This is a comprehensive review of the clinical aspects of migraine in the elderly and on the rationale for using ketogenic diet as a treatment in migraine.

I have 2 major comments and suggestions.

Comment: 1) in § 5 the authors cite only reviews on ketogenic diet in migraine (not mentioning for example the 1st RCT by Di Lorenzo's group) and at the same time underline the heterogeneity of dietary programs. This is obviously important for the effects and side effects of ketogenic diets. I would suggest a table mentioning the major controlled trials underlining their difference in composition.

Answer: The major clinical trials with the detailed specification of dietary program have been just presented in a table in a recent review of Neri et al., (doi10.3389/fnut.2023.1204700) and therefore we do not want to repeat the information that has just been published. We have added the following fragment to the last paragraph in the 5. Section:

“Several randomized clinical trials evaluating the role of ketogenic intervention in migraine were performed. The first such trail of Di Lorenzo et al. included overweighted migraine patients during a weight-loss intervention [Di Lorenzo]. Therefore, these results cannot be directly related to this review as (1) migraine and obesity is an important and autonomous problem; (2) overweight/obesity in the elderly is a problem, but it is somehow overwhelmed by unexplained weight loss, typical for this group of patients. A recent review of Neri et al., presents clinical trials and other studies in migraine patients with ketogenic intervention resulting in production of endogenous ketone bodies or administration of exogenous ketone bodies [74]. […] So far, no clinical trials with ketogenic intervention in elderly migraine patients have been performed.”

with new reference:

Di Lorenzo, C.; Pinto, A.; Ienca, R.; Coppola, G.; Sirianni, G.; Di Lorenzo, G.; Parisi, V.; Serrao, M.; Spagnoli, A.; Vestri, A.; et al. A Randomized Double-Blind, Cross-Over Trial of very Low-Calorie Diet in Overweight Migraine Patients: A Possible Role for Ketones? Nutrients 2019, 11, doi:10.3390/nu11081742.

Comment: Moreover in § 5 (but also elsewhere) the reference numbers are not adequate: for instance ref 73 and 74 should be 74 and 75. I recommend to check all references in the manuscript.

Answer: We have reordered in-text citations.

Comment: 2) in § 6 the authors discuss the potential harm ketosis could produce in the elderly. This section is difficult to understand because it is a mixture of experimental data in animals and clinical human data. These 2 data sets should be clearly separated and again it would be instructive and reader-friendly to have a table showing the most prominent animal data on the one hand, and clinical studies on the other. At least the latter should be summarized in a table as there seem to be many contradictory results in the literature (as also mentioned by the authors)

Answer: We have rewritten the paragraphs that might be unclear in their original version, but to our knowledge, the problem of potential harm of ketone bodies was experimentally/clinically addressed only in 3 papers (2 for humans and 1 for mice), so we do not present their results in tables as they are described in detail in revised text. Moreover, as we stated above, there is a recent review presenting detailed information on clinical trials and other studies and our review should rather update information and not copy recent information.

We have changed the fragment:

“Moreover, a recent study reported a dose-response relationship of 50% increase in all-cause mortality between lowest and highest quintiles of ketone body concentrations in White and Black Americans [81]. Moreover, the levels of ketone bodies were associated with incident heart failure and higher all-cause mortality. These associations suggest a potential detrimental effect of ketone body metabolism in aging.

To explore mechanism underlying the role of ketogenesis in mammalian life span, the Hmgcs2-/- mice, incapable of endogenous ketogenesis, were generated [82]. 3-hydroxy-3-methylglutaryl-CoA synthase 2 (HMGCS2) is a mitochondrial enzyme critical for the second stage of ketogenesis [83]. Using the Hmgcs2-/- mice, their normal Hmgcs2+/+ counterpart, and applying a normal diet, a ketogenic diet (4.5% carbohydrate, 80.8% fat, and 14.7% protein) and a normal diet enriched with 1,3-butanediol, a precur-sor of -hydroxybutyrate showed that ketone bodies might have significant effect on life span of animals in two periods of life: just after the birth and older age. Specifically, mice on ketogenic diet displayed a higher mortality in old age and mice on diet supplemented with 1,3-butanediol exhibited higher mid-life mortality, but their mortality in old age was like control animals. Therefore, it was hypothesized that supple-mentation with ketone bodies just prior to old age might extend life span. This hypothesis was positively verified, but it was also shown that the intervention with keto-genic diet at the same age shortened life span [84].

A positive correlation between circulating ketone bodies and all-cause mortality was observed in the general population-based cohort study Prevention of Renal and Vascular Endstage Disease Intervention (PREVEND) of individuals aged 54 ± 12 years [85]. Moreover, this study also showed that an increased all-cause mortality associated with an increased fatty liver index, a proxy of nonalcoholic fatty liver disease, was partly mediated by circulating ketone bodies.”

into:

“Two human studies reported a harmful effect of ketone bodies. A positive correlation between circulating ketone bodies and all-cause mortality was observed in the general population-based cohort study Prevention of Renal and Vascular Endstage Disease Intervention (PREVEND) in individuals aged 54 ± 12 years [85]. This study also showed that an increased all-cause mortality was associated with an increased fatty liver index, a proxy of nonalcoholic fatty liver disease and this increase was partly mediated by circulating ketone bodies. A recent study reported a dose-response relationship of 50% increase in all-cause mortality between lowest and highest quintiles of ketone body concentrations in White and Black Americans [81]. Moreover, the levels of ketone bodies were associated with incident heart failure and higher all-cause mortality. These associations suggest a potential detrimental effect of ketone body metabolism in aging.

The mechanism behind these observations in human is partly explained in an experiment with homozygotic mice with a mutation in the 3-hydroxy-3-methylglutaryl-CoA synthase 2 (HMGCS2) gene encoding for the protein critical for the second stage of ketogenesis [82]. These animals are incapable of endogenous ketogenesis. The HMGCS2-/- mice were given a normal diet, a ketogenic diet (4.5% carbohydrate, 80.8% fat, and 14.7% protein) and a normal diet supplemented with 1,3-butanediol, a precursor of b-hydroxybutyrate showed that ketone bodies might exert a significant effect on life span of animals in two periods of life: just after the birth and older age. Specifically, mice on ketogenic diet displayed a higher mortality in old age and mice on diet supplemented with 1,3-butanediol exhibited higher mid-life mortality, but their mortality in old age was like control animals. An ad libitum ketogenic diet increased mortality, suggesting that not only proportion of the main components of the ketogenic, but also its rate of administration may decide about its outcome.”

Comment: The abstract contrast with §6, because it only mentions the harm ketogenic diet may cause, and not the fact that results are contradictory and that the conclusion that "ketogenic diet should not be recommended in older adults with migraine" could be somewhat nuanced.

Answer: We have changed the following fragment of the abstract:

“It may force organisms to switch from glucose to ketone bodies as the primary source of energy production. In general, the ketogenic diet and the action of ketone bodies are considered beneficial for several aspects of health. Studies on the ketogenic diet in migraine are not standardized and poorly evidenced. Recent studies within the Cardiovascular Health Study and Prevention of Renal and Vascular Endstage Disease Intervention, showed that increased levels of ketone bodies, might be associated with all-cause and incident heart failure mortality in older adults. These results were supported by studies on mice showing that the ketogenic diets and diet supple-mentation with a precursor of b-hydroxybutyrate, the main human ketone body, might cause life span shortening. These facts support the thesis that the ketogenic diet should not be recommended in older adults with migraine.”

into:

“It may replace glucose by ketone bodies as the primary source of energy production. The ketogenic diet and the action of ketone bodies are considered beneficial in several aspects of health, including migraine prevention , but studies on the ketogenic diet in migraine are not standardized and poorly evidenced. Apart from papers claiming beneficial effects of ketogenic diet in migraine, some studies report that increased levels of ketone bodies may be associated with all-cause and incident heart failure mortality in older adults and are supported by research on mice showing that the ketogenic diets and diet supplementation with a human ketone body precursor, might cause life span shortening. Therefore, despite reports showing a beneficial effect of the ketogenic diet in migraine, such diet requires further studies, including clinical trials, to verify whether it should be recommended in older adults with migraine.”

Reviewer 2 Report

Comments and Suggestions for Authors

The topic of literature review is novel, the content of the review is comprehensive, there are some problems, please revise

1, the title: the title uses a question mark, under normal circumstances, the current existing views or technology can be questioned with a question mark, if the ketogenic diet is not widely standardized for the clinical treatment of migraine, then the question mark is inappropriate.

2. Introduction: The introduction should explain why this review is carried out, outline its significance and application prospects.

3. As can be seen from Figure 1, the incidence of migraine in women is much higher than that in men. Please analyze the possible causes and mechanisms.

4, 2. Migraine and Its Relation to Age and Nutrition: This section should focus on the ketogenic diet related nutrients and migraine.

5. Although Figure 2 is beautiful, it shows little information about the research results, and the author needs to make reasonable modifications.

6. The information shown in Figure 3 belongs to common sense in the field of nutrition, and the figure of literature review should show new ideas rather than common sense.

Author Response

Reviewer #2

The topic of literature review is novel, the content of the review is comprehensive, there are some problems, please revise

Zmienić: Finally, we present arguments against the application of the ketogenic diet in the el-derly suffering from migraine.

Comment: 1, the title: the title uses a question mark, under normal circumstances, the current existing views or technology can be questioned with a question mark, if the ketogenic diet is not widely standardized for the clinical treatment of migraine, then the question mark is inappropriate.

Answer: We have changed the title into:

“The Ketogenic Diet in the Prevention of Migraine in the Elderly”

Comment: 2. Introduction: The introduction should explain why this review is carried out, outline its significance and application prospects.

Answer: We have added the following paragraph to the Introduction section:

“Migraine is one of the most serious reasons of disability and affect a significant part of older adults population, adding another health concern in this group of individuals. The ketogenic diet is recommended in migraine prevention, but there is not a clinical trial evaluating its efficacy and safety in elderly migraine sufferers. This review has been written to provide information and arguments to verify whether the ketogenic diet should be applied in the prevention of migraine in the elderly. It is obvious that the final conclusion must be verified empirically, through pre- and clinical studies.”

Comment: 3. As can be seen from Figure 1, the incidence of migraine in women is much higher than that in men. Please analyze the possible causes and mechanisms.

Comment: As we wrote “Although the exact reason for this diversity is not known, several possible mechanisms are considered to underline this difference, including sex hormones, pregnancy, the menstrual cycle, X-linked form of migraine and mitochondrial transmission [18,19]”

and we think that this enough for this review, especially that the difference between migraine prevalence in women and men decreases with age and in the elderly is about a few percent. The consideration of possible causes and mechanisms of the difference in migraine prevalence between women and men may be a subject of another review.

Comment: 4, 2. Migraine and Its Relation to Age and Nutrition: This section should focus on the ketogenic diet related nutrients and migraine.

Answer: The 2. Migraine and Its Relation to Age and Nutrition section is rather introductory section showing the role of nutrition in migraine pathogenesis. Ketogenic diet-related nutrients are discussed in the section 5. The Ketogenic Diet and Ketone Bodies in Migraine.

Comment: 5. Although Figure 2 is beautiful, it shows little information about the research results, and the author needs to make reasonable modifications.

Answer: It was not our intention to present any research results in Figure 2. We wanted to show typical features of migraine in the elderly. Of course, the picture within the figure brings little information, if any, and we are ready to replace this figure with a table.

Comment: 6. The information shown in Figure 3 belongs to common sense in the field of nutrition, and the figure of literature review should show new ideas rather than common sense.

Answer: Indeed! We have removed this figure.

Round 2

Reviewer 1 Report

Comments and Suggestions for Authors

I'm satisfied with the modifications of this revised version.

Reviewer 2 Report

Comments and Suggestions for Authors

 Accept in present form